# Molecular *BCR::ABL1* Quantification and *ABL1* Mutation Detection as Essential Tools for the Clinical Management of Chronic Myeloid Leukemia Patients: Results from a Brazilian Single-Center Study

**DOI:** 10.3390/ijms241210118

**Published:** 2023-06-14

**Authors:** Anelis Maria Marin, Denise Kusma Wosniaki, Heloisa Bruna Soligo Sanchuki, Eduardo Cilião Munhoz, Jeanine Marie Nardin, Gabriela Silva Soares, Dhienifer Caroline Espinace, João Samuel de Holanda Farias, Bruna Veroneze, Luiz Felipe Becker, Guilherme Lima Costa, Olair Carlos Beltrame, Jaqueline Carvalho de Oliveira, Geison Cambri, Dalila Luciola Zanette, Mateus Nóbrega Aoki

**Affiliations:** 1Laboratory for Applied Science and Technology in Health, Carlos Chagas Institute, Oswaldo Cruz Foundation (Fiocruz), Curitiba 81350-010, Brazil; anelis.marin@fiocruz.br (A.M.M.); denisewosniaki@gmail.com (D.K.W.); helosoligo@gmail.com (H.B.S.S.); gabriela.soares@fiocruz.br (G.S.S.); dhienifercaroline@hotmail.com (D.C.E.); 2Erasto Gaertner Hospital, Curitiba 81520-060, Brazil; edciliao@hotmail.com (E.C.M.); jeaninemarie.nardin@gmail.com (J.M.N.); jshfarias@erastogaertner.com.br (J.S.d.H.F.); brunaveroneze@gmail.com (B.V.); lfbecker9@gmail.com (L.F.B.); guilhermelimacosta@hotmail.com (G.L.C.); ocbeltrame@gmail.com (O.C.B.); 3Department of Genetics, Federal University of Parana, Curitiba 81531-980, Brazil; jaqueline.carvalho@ufpr.br; 4Instituto de Biologia Molecular do Paraná (IBMP), Curitiba 81350-010, Brazil; geison.cambri@gmail.com

**Keywords:** chronic myeloid leukemia, *BCR::ABL1*, *ABL1* mutation, TKI, prognostic

## Abstract

Chronic myeloid leukemia (CML) is a well-characterized oncological disease in which virtually all patients possess a translocation (9;22) that generates the tyrosine kinase *BCR::ABL1* protein. This translocation represents one of the milestones in molecular oncology in terms of both diagnostic and prognostic evaluations. The molecular detection of the *BCR::ABL1* transcription is a required factor for CML diagnosis, and its molecular quantification is essential for assessing treatment options and clinical approaches. In the CML molecular context, point mutations on the *ABL1* gene are also a challenge for clinical guidelines because several mutations are responsible for tyrosine kinase inhibitor resistance, indicating that a change may be necessary in the treatment protocol. So far, the European LeukemiaNet and the National Comprehensive Cancer Network (NCCN) have presented international guidelines on CML molecular approaches, especially those related to *BCR::ABL1* expression. In this study, we show almost three years’ worth of data regarding the clinical treatment of CML patients at the Erasto Gaertner Hospital, Curitiba, Brazil. These data primarily comprise 155 patients and 532 clinical samples. *BCR::ABL1* quantification by a duplex-one-step RT-qPCR and *ABL1* mutations detection were conducted. Furthermore, digital PCR for both *BCR::ABL1* expression and *ABL1* mutations were conducted in a sub-cohort. This manuscript describes and discusses the clinical importance and relevance of molecular biology testing in Brazilian CML patients, demonstrating its cost-effectiveness.

## 1. Introduction

Chronic myeloid leukemia (CML) is characterized by the presence of the Philadelphia chromosome (Ph) resulting from the reciprocal translocation t(9;22) (q34;q11). This translocation gives rise to the *BCR* (breakpoint cluster region protein) *ABL1* (Abelson murine leukemia viral oncogene homolog 1) fusion gene, which produces a constitutively activeyrosine-kinase, resulting in the general imbalance found in CML [1]. CML represents 15% of adult leukemias, and, in 2023, the American Cancer Society estimated that it caused about 8930 new cases and 1310 deaths per year in the United States (https://cancerstatisticscenter.cancer.org/#!/cancer-site/Leukemia, accessed on 1 March 2023). In Brazil, the INCA (National Institute of Cancer) estimated that 10,070 new leukemia cases were diagnosed in 2017, of which 10% were CML [1]. The current therapy is based on tyrosine-kinase inhibitors (TKIs) such as imatinib, dasatinib, nilotinib, bosutinib, and ponatinib. In Brazil, the Unified Health System (SUS) provides recommendations on diagnosis and treatment in its clinical protocols and therapeutic guidelines. These guidelines were prepared by the CONITEC (National Commission for the Incorporation of Technologies) and are based on European LeukemiaNet recommendations [2].

The largest oncology hospital in the state of Paraná, the Erasto Gaertner Hospital, has a vast center of Hematology and Hemotherapy, which performs therapies of the highest complexity, such as Cellular Therapy with T Cells (CAR T), in addition to Bone Marrow Transplantation for various oncohematology pathologies. The majority of Erasto Gaertner Hospital patients (80%) are assisted by SUS, so data from this hospital is adequate to show the relevance and prevalence of CML, as well as the impact of the introduction of TKIs in the early 2000s. In a survey carried out using audit data from Authorizations for Outpatient Procedures (APAC), there were more than 4800 TKIs emissions between January 2008 and January 2023. A total of 75% of the emissions were for imatinibe, and approximately 35% were for second-generation inhibitors (dasatinib and nilotinib), corroborating the international literature, which presents approximately 1/3 of patients as refractory to the first line of therapy [3,4]. When the CML response fails to 2GTKIs, the use of allogeneic bone marrow transplantation is suggested [5,6]. In the Erasto Gaertner Hospital, of the 158 bone marrow transplants performed in 2021, 2 were for CML, and in 2022, 5 were for CML patients. So, even with the noticeable response rate of TKIs, a small percentage of the total transplants performed are reserved for this condition.

CML is divided into three phases, which are distinguished by the number of immature blasts in the blood or blood marrow: chronic (CP), accelerated (AC), and blast phase (BP) [7]. Diagnosis usually occurs during the chronic phase, which can progress into the accelerated phase if not treated [8]. The National Comprehensive Cancer Network (NCCN) guidelines discuss the management of CML in all three phases [9]. According to the Sokal and Hasford scoring systems, patients in the chronic phase can be classified into three risk groups (low, intermediate, and high). Patients at low risk are first submitted to treatment with 1GTKIs such as imatinib, and if no response is obtained, they are treated with second-generation TKIs. However, intermediate- or high-risk patients are directly submitted to treatment with second-generation TKIs [9]. The diagnosis includes palpation of the spleen, complete blood count, bone marrow aspiration, and biopsy for morphological and cytogenetic evaluation, in addition to RT-qPCR to establish the presence of *BCR::ABL1* mRNA [10].

RT-qPCR is usually performed with cells from the peripheral blood during the initial workflow and can be correlated to the number of CML cells and measured using an international scale (IS) of molecular response [11]. In CML patients, the *BCR::ABL1* quantification in peripheral blood cells at the time of diagnosis is set as 100%, and the subsequent quantifications are expressed in percentage by the *BCR::ABL1/ABL1* ratio. The reduction in this ratio in subsequent tests is used to assess the expected response to treatment. When there is a reduction of 2 logs, representing 1% of the original level, there is a complete cytogenetic response (CCyR). Likewise, a reduction of 3 logs (0.1% of tumor cells) is referred to as a major molecular response (MMR or MR^3^), and a reduction of 4 logs is called deep molecular response (DMR—MR^4^, MR^4.5^, MR^5^) [12].

The recommended procedures for treatment follow-up include the quantification of *BCR::ABL1* transcripts every three months in the first year of treatment and then every six months once MMR is reached. However, if the quantification of *BCR::ABL1* remains ≥10% after 3–6 months of treatment, or ≥1% after 12 months, the treatment is considered to have failed. Both cases are indicative of resistance mutations or a high risk of abnormal chromosome aberration in Ph^+^. In patients who do not reach DMR, treatment may be changed to a second-generation TKI (2GTKI) to improve the depth of response [13].

On the other hand, MMR at 12 months is associated with a lower rate of disease progression and a higher probability of reaching DMR—a prerequisite for discontinuation of the treatment, which, when successful, is called treatment-free remission (TFR). To try discontinuation protocol, a minimum 5 years of TKI therapy with a sustained DMR of at least three years in MR^4^ and two years in MR^4.5^ is required [13], confirming the importance of monitoring *BCR::ABL1* levels.

Imatinib belongs to the first generation of TKIs (1GTKIs), which substituted the previous treatment with recombinant interferon alpha (IFNα) and low-dose cytarabine. The IRIS study showed higher rates of cytogenetic and molecular responses and better progression-free survival (PFS) and overall survival (OS) in patients treated with imatinib [14,15]. Between 60 and 80% of patients reach early molecular response (*BCR::ABL1* ≤10% IS) in three or six months; 20–59% reach MMR within one year; 60–80% reach MMR in five years; and 35–68% reach DMR in five years [11]. PFS rates at five years are 80–90%, and OS rates are 90–95%, while OS at 10 years ranges from 82 to 85% with a death rate of 6% [14,16]. Other TKIs, called second-generation TKIs (2GTKI), such as dasatinib, nilotinib, and bosutinib, possess chemical structure modifications that make them more potent against the *BCR::ABL1* tyrosine kinase, reducing the side effects. Imatinib resistance occurs in 10–15% of cases and in about 10% of patients treated with 2GTKIs. Known mechanisms of resistance include clonal evolution, mutations in the *BCR::ABL1* kinase domain, and activation of *BCR::ABL1* independent pathways. Many *BCR::ABL1* kinase domain mutations have been described. Changes in treatment may be recommended depending on the mutation [13,17,18], as patients harboring T315I mutation are advised to use Ponatibib, while V299L mutation requires the use of Nilotinib or ponatinib and Y253H, E255V/K, F359V/I/C Dasatinib, bosutinib, or ponatinib. Patients with TKI-resistant diseases are evaluated for changes from 1G to 2G or 3G TKI treatment or for allogeneic hematopoietic cell transplantation [8].

Dasatinib has a 325-fold higher potency in inhibitory activity against some *ABL1* kinase domain resistance mutations [18]. In addition, a DASISION trial found better results in patients treated with dasatinib than imatinib [19]. They found significant increases in early molecular response (84%), MMR rate after one year (46%), and the five-year probability of achieving MMR (76%) and MR^4.5^ (42%). However, PFS and OS were no higher than imatinib (86% vs. 85% and 91% vs. 95%, respectively).

Nilotinib, another 2GTKI, was developed to have enhanced selectivity and potency toward *BCR::ABL1* tyrosine kinase and also demonstrates clinical activity against some *ABL1* mutations. It is recommended as a second-line therapy for CP or AC patients resistant or intolerant to imatinib [18]. The drug showed promising results in the ENESTnd trial [20]. The five- and ten-year probabilities of achieving MMR were better than with imatinib (77% and 82.6%), MR^4^ (66% and 73%), and MR^4.5^ (54% and 64%, respectively). The five- and 10-year OS, meanwhile, were not different from imatinib (94% vs. 92% and 87.6% vs. 88.3%, respectively).

Bosutinib is a 3G TKI that also inhibits several *BCR::ABL1* resistance mutants. The BFORE study found that the early molecular response rate (75%) and one-year MMR rate (47%) with bosutinib were significantly higher than with imatinib [21]. Ponatinib is another 3G TKI that is more potent than any other TKI and is indicated for patients with the T315I mutation and for patients with resistance to two or more TKIs [22]. The resistance mechanism involved in T315I mutation relies on the oscillation between active and inactive states of the enzyme due to their auto-phosphorylation activity. The threonine residue at the 315 position is one of the important sites for phosphorylation, and when it changes to isoleucine, phosphorylation does not occur. As the conformational change necessary for most TKIs to interact with the enzyme does not happen, the result is resistance to imatinib, dasatinib, nilotinib, and bosutinib [23].

Several clinical studies have evaluated the effectiveness of treating CML patients with different TKIs [14,19,24,25,26,27,28], with the most recent ones indicating that 64% of patients treated with imatinib after 5–10 years reach MMR status. For the patients that do not respond to treatment, various TKI replacements are indicated. Therefore, sensitive molecular monitoring *BCR::ABL1* is essential for all patients [29,30,31].

Imatinib treatment for CML patients is quite costly, even with the availability of generics to Gleevec^®^ [32,33]. To illustrate the cost of CML treatment in a Brazilian context, consider an average rate of 1621 new cases of CML every year and a 10-year survival rate of 90%, which is generally in line with existing studies of the disease [14,21,28,34]. If we only consider cases that emerged in the last 10 years, there are 14,589 CML patients under treatment in Brazil. Given that 66% of patients treated with imatinib achieve MMR after five years, treatment is ineffective for one-third of patients, i.e., 4960 patients. According to Valor Brasindice, Gleevec^®^ 400 mg (Novartis, Basel, Switzerland) is priced at around $3870, while generic imatinib is priced at $2515. This amount can be reduced with a subsidy; therefore, considering a positive scenario with a partial subsidy for generic imatinib, the monthly cost per patient is approximately $1200. The cost of treating these 4960 CML patients who do not respond to imatinib therefore represents an annual expenditure of over $70 million on an ineffective treatment. Regarding the other group comprised of around 8150 patients treated with imatinib for more than five years, 66% (5187 patients) will achieve MMR and DMR and can be included in the TFR protocol. Studies show that only 35% of patients in TFR will have a recurrence in six months [35]. Therefore, considering that 65% of the 5187 patients included in TFR will not have a recurrence (3372 patients) at a monthly cost of $1200 per patient, the safe suspension of the drug in this group would represent an annual saving of almost $48,000,000.

CML is characterized by a period of innate and adaptive immune dysfunction at diagnosis, which prevents antileukemia immune responses. Immune suppressor cells, including myeloid-derived suppressor cells (MDSCs) and regulatory T-cells (Tregs), contribute to T-cell dysfunction and disease progression in CML, expanding at diagnosis, and reducing following TKI therapy. Clinical data showed that imatinib or dasatinib-treated patients exhibit an expansion of CD8+ CTLs or NK cells, which are associated with an improved response to therapy. Concerning immunotherapies for CML, IFN-α may provide an alternative approach in selected TKI-resistant mutation-positive CML patients, particularly those ineligibles for bone marrow transplantation. A combined therapy with Chimeric Antigen Receptor T-cells therapy (CAR-T) and TKI may confer particular therapeutic benefit to TKI-resistant/intolerant, young, or advanced phase CML patients [36].

In this manuscript, we describe an almost three-year molecular oncology project in CML patients in a single Brazilian treatment center, with several approaches that provide clinically relevant and directly applied data, resulting in positive public health and financial impacts in a complex and expansive scenario.

## 2. Results

### 2.1. Characterization of Duplex One-Step RT-qPCR for BCR::ABL1 Quantification Assay

Standard curves were used to calculate the limit of quantification (LoQ) for both *BCR::ABL1* and *ABL1*. LoQ represents the lowest detected concentration that maintains the linear regression with an r^2^ equal to or higher than 0.99, ensuring that data extrapolation enters the direct quantification interval. We observed an LoQ of 500 copies/well for both *BCR::ABL1* and *ABL1* (Figure 1). It is important to state that, from 18 wells with 5 × 10^2^ of *BCR::ABL1* and *ABL1*, only two and one were not amplified, respectively.

### 2.2. Amplification Efficiency and Repeatability

Figure 1 and Table 1 shows the mean cycle threshold, standard deviation, and repeatability parameters. Figure 2 depicts the graphic representation, showing an amplification efficiency of 90% for *BCR::ABL1* and 87% for *ABL1*.

### 2.3. Cross-Validation, Specificity, and Sensitivity

The cross-validation assay was performed by comparing the results obtained from our test with those obtained from a private laboratory. The results for 24 samples were similar, although a slight difference was observed in *BCR::ABL1* detection and quantification, as demonstrated in Table 2.

Analytical specificity and sensitivity were assessed using RNA from 16 human cell lines, of which 14 were negative and 2 were positive t(9;22). The assay had a specificity and sensitivity of 100%, with none of the negative cells exhibiting amplification of the *BCR::ABL1*, while all two positive cell lines showed amplification.

When *BCR::ABL1* of 19 samples were quantified in parallel with DualQuant and Mobius commercial kit, we observed that DualQuant assay provided a significantly lower cycle threshold compared with the commercial kit (Table 3).

### 2.4. Assay Validation and Certification

Finally, using the secondary reference calibrator panel DualQuant assay presents a calibration factor of 0.10—a value that must be used for *BCR::ABL1* quantification on the International Scale. Moreover, it was clear that our assay had a sensitivity of 0.001% of BCR-ABL in clinical samples, which corresponds to MR^4.5^—achieving deep molecular response.

### 2.5. Epidemiological Data of CML Samples

During the 33 months of follow-up, 155 CML patients were recruited, generating a total of 532 clinical samples. From this cohort, 86 patients (55.4%) were male and 69 (44.6%) were female. The medium age at diagnosis was 50.56 (±14.94) years old. The number of samples per patient varies from 1 to 15 due to several factors such as disease stage, treatment response, recommended follow-up milestones, and clinical outcomes. Figure 2 shows 45, 19, 18, and 8 patients with three, four, five, and six clinical samples, respectively. In the treatment context, 67 of 155 patients (43.2%) had to change from imatinib to another TKI, mainly dasatinib and nilotinib.

### 2.6. Response to Treatment Evaluated by Quantification of BCR::ABL1 Expression

In this clinical cohort, we performed *BCR::ABL1* quantification by RT-qPCR for 496 samples obtained from the 155 CML patients. Molecular quantification of *BCR::ABL1* aims to evaluate treatment efficacy and adaptation and reveals the undeniable value of this molecular tool for clinical decision and management. Considering *BCR::ABL1* molecular monitoring, our cohort was divided into four groups: (1) *BCR::ABL1* reduction; (2) Drug alteration; (3) Persistently low *BCR::ABL1* patients; (4) Allogeneic Hematopoietic Stem Cell Transplantation (allo-HSCT). The first group comprises patients presenting a reduction in *BCR::ABL1* levels during follow-up, demonstrating the effectiveness of TKI treatment. The second group comprises the patients presenting an increase in molecular *BCR::ABL1* levels, demonstrating that TKI treatment is not effective and that there is a need to change the treatment protocol. The third group comprises patients with molecular *BCR::ABL1* quantification lower than 1% but never reaching MMR 4.0, showing persistently low levels. This feature may indicate the maintenance of the leukemic clone. The fourth group comprises patients that underwent BMT, either because of resistance or relapse. Figure 3 shows the dynamics of *BCR::ABL1* molecular levels for these different groups of patients.

Allogeneic hematopoietic stem cell transplantation (allo-HSCT) is recommended in CML when disease resistance occurs, including suboptimal response to two or more TKIs or even failure to respond to Ponatinib. It is crucial to monitor *BCR::ABL1* levels to detect relapse after allo-HSCT as early as possible. In this cohort, five patients underwent allo-HSCT. Three of them did not show any of the TKI resistance mutations analyzed here (T315I, E255K, Y253H, V359F), while two of them presented the T315I mutation. From the three wild-type cases that underwent HSCT, two of them are currently stable and are not being treated with TKIs, while one case presented severe GVHD, the major complication after HSCT. The other two HSCT patients carried the T315I mutation. One of the T315I cases currently has undetectable levels of *BCR::ABL1* and is under Dasatinib treatment. Importantly, the T315I mutation was no longer detected after allo-HSCT in this patient. The other T315I case is under Ponatinib treatment but showed a worse response, with the most recently measured levels of *BCR::ABL1* being greater than 1%.

The number and percentages of patients in each level of response to TKIs can be seen in Table 4—divided by each year of follow-up (2020, 2021, 2022, 2023). The table refers to the latest quantification of *BCR::ABL1* IS% of each year for each unique patient. The number and percentage of patients that reached DMR in each year are also shown and correspond to the sum of patients with MMR of 4.0 or less. The average percentage of patients that reached DMR in each complete year of follow-up was 49.1% (for 2020, 2021, and 2022).

The total number of unique patients that reached DMR in the whole 33 months of follow-up was obtained by selecting only the latest sample in the total period for each patient. By doing so, from a total of 178 unique patients, 99 (55.6%) reached DMR, while the remaining patients showed different levels of response, as shown in Table 5. The vast majority of patients (47.8%) showed undetectable levels of *BCR::ABL1* IS% in their last visit to the hospital.

TKI discontinuation on CML patients—molecular *BCR::ABL1* quantification is an essential requirement for the treatment-free remission protocol, as patients must be on TKI with DMR for two years to be eligible. We subdivided the patients in our study cohort receiving imatinib treatment into those who are within TFR requirements for one- and two-year follow-ups. In total, we found 14 patients in DMR or undetected *BCR::ABL1* expression at their one-year follow-up, indicating that they should have one more year of molecular quantification to have TFR requirements. In addition, 19 patients are currently under TFR requirements and eligible for imatinib discontinuation. In our cohort, there are six CML patients in the TFR protocol, with imatinib discontinuation ranging from 294 to 602 days (440.33 ± 98.91 days), all of whom are in DRM or with undetected *BCR::ABL1* expression for all clinical samples quantified by RT-qPCR (Figure 4). We highlighted two CML patients where six and nine clinical samples had *BCR::ABL1* quantification, indicating the effectiveness of the TFR protocol.

To confront the qPCR data, digital PCR (dPCR) was performed for the most recently collected samples from 22 patients in DMR. These patients had DMR or undetected *BCR::ABL1* expression at their one-year follow-up or are under TFR requirements and eligible for imatinib discontinuation. Table 6 shows the results of dPCR and the *BCR::ABL1* IS% obtained by qPCR, showing that nine patients had undetectable levels of BCR::ABL by qPCR, while dPCR showed detection of copies. Two patients showed BCR::ABL quantification by qPCR but no detection in dPCR. Figure 5 illustrates the dPCR analysis of three samples that were also represented in Table 6.

### 2.7. ABL1 Mutation

For this, 373 assays were performed on 143 patients for the detection of the five variations mentioned previously to examine the presence of the *ABL1* mutation in CML patients by qPCR. T315I (rs121913459) was detected in four patients, E255K (rs121913448) was detected in one patient, and V359F (RS121913452) was detected in one patient. T315I mutation results in clinical resistance to first- and second-line TKIs such as imatinib, dasatinib, and nilotinib; hence, ponatinib is indicated in these cases. The E255K mutation confers a low response to all TKIs, including ponatinib. The V359F mutation is clinically linked with a positive correlation with dasatinib, which is the most effective treatment for patients with this mutation. We found two patients who carry the T315I mutation, who presented 43.8% and 80.4% of *BCR::ABL1* expression by molecular quantification, and were submitted to allo-HSCT. After allo-HSCT, for these two patients, the peripheral blood was screened for rs121913459 (T315I) by qPCR and showed no presence, indicating an effective transplantation and highlighting the importance of screening *ABL1* mutations. However, in posterior samples from these patients, *BCR::ABL1* increased again, and dPCR analysis showed the presence of T315I (Table 7—Patient 2 and 3), suggesting that dPCR can be a useful method for earlier mutation detection.

Samples were chosen for dPCR to confirm the qPCR results and find divergences that can be used to explain resistance profiles. Besides the two cases of allo-HSCT, Table 8 shows that three patients (4, 5, and 6) were refractory to the treatments and had no mutations detected by qPCR; however, dPCR was able to detect T315I mutation. However, for Patient 1 and Patient 8, no difference was observed between qPCR and dPCR. Perhaps this difference could have been detected in a time point between the two samples analyzed here, which would allow the earlier detection of the mutation. In Sample 2 for Patient 7, the E255K mutation could be detected by dPCR but not by qPCR, and after the change in TKI treatment, the *BCR::ABL1* decreased, showing that this patient is currently responding to treatment, even though this mutation showed poor response to dasatinib [12]. Together, these results strongly suggest that it is possible to monitor the emerging mutations by dPCR, improving the prognosis and choice of TKI. Figure 6 represents the analysis of mutations by dPCR. The T315I mutation is given by C > T substitution, then the assay purchased from ThermoFisher contains one probe with T (green channel) and one with C (yellow channel).

### 2.8. Cost-Effectiveness

Cost-effectiveness analysis considers the sub-cohort that is eligible for TKI discontinuation. As previously described, 19 patients are currently eligible for the TFR protocol, while 14 will become eligible in one year. Based on the simplifying assumption that around 65% of TFR patients will be successful in the TFR protocol [35], our cohort is able to immediately include twelve patients in the TFR protocol (group 1) and nine in one year (group 2). Considering the monthly cost of imatinib of $1200/patient, i.e., $14,400 per year/patient, the TFR protocol will save around $177,840 for group 1 in the first year and $131,000 for group 2 in the second year. Furthermore, in the second year, the annual savings for the sum of groups 1 and 2, representing 21 patients, will amount to $302,400, as demonstrated in Figure 7 and Table 8. More importantly, the savings will improve over time as new CML patients that are eligible for TFR will be included in the TFR protocol. In a simulation for HEG patients, if eight novel patients are included annually in the TFR protocol, after five years, the savings will be around $650,000.00, considering that CML is a chronic disease with long survival rates.

The cost of RT-qPCR quantification of *BCR::ABL1* is $900 per patient per year, considering a unitary test value of $100 and nine tests/per year per patient. So, in a simplified way, in the first year of the TFR protocol, $14,400 of imatinib costs could be replaced by a cost of $900 with RT-qPCR per patient.

## 3. Discussion

The current clinical management and guidelines of CML treatment require the close molecular monitoring of *BCR::ABL1* expression for both treatment effectiveness evaluation guiding to targeted therapy and TKIs discontinuation protocol. These approaches are primarily concerned with proper and effective therapy and closely correlated with financial issues. Several in-house RT-qPCR *BCR::ABL1* quantification assays are available; however, they present substantial variation, thus hindering inter-laboratory discrepancies [37,38,39]. This led to the establishment of an international scale (IS) for *BCR::ABL1* quantification, which resulted in a laboratory-specific conversion factor (CF). For Brazilian and Latin American laboratories, a secondary reference calibrators panel was developed by Bianchini et al. to provide a harmonization panel for *BCR::ABL1* measurement that could be applied to the molecular monitoring response of patients [40].

Effective TKI therapy in CML patients requires the achievement of a deep molecular response (MR^4^, MR^4,5^, and MR^5^) [41,42]. A Brazilian CML experts group has highlighted the need for molecular monitoring of *BCR::ABL1* in the Brazilian Unified Health System (SUS), noting that this approach should be part of the integral treatment of patients with CML since this monitoring will reduce the chances of disease progression, decrease health system costs, ensure compliance with international guidelines, and allow eligible patients to enter the TFR protocol, which will lead to cost savings that more than offset the cost of molecular testing [43]. One Brazilian study has reported the results of *BCR::ABL1* molecular monitoring in CML patients, with 60 patients followed from June 2005 until September 2008, with hematological, major cytogenetic, and complete cytogenetic responses achieved by 95%, 75%, and 63% patients, respectively, while 40% of patients achieved a major molecular response in a median time of 8.5 months [44]. In another study, 1117 CML patients treated with TKI therapy for more than two years were assessed through 3373 peripheral blood samples in a European survey, which found that 22.64%, 30.98%, and 23.47% reached MR^4^, MR^4,5^, and MR^5^, respectively, indicating that more than 77% of CML patients in the cohort had a good clinical and molecular response [45]. When *BCR::ABL1* quantification was compared in 631 paired peripheral blood and bone marrow samples from 283 CML patients, a good overall concordance was observed, but there was a systematic tendency towards higher *BCR::ABL1* levels in peripheral blood than in bone marrow, supporting the current practice of using peripheral blood [46]. It was demonstrated that patients who achieved sustained MR^4^, i.e., *BCR::ABL1* RT-qPCR < 0.01% IS for 12 months, showed a negligible risk of regressing to MR^3^, suggesting that MR^4^ can be characterized as a secure molecular threshold and that patients within this group would need less frequent monitoring [36]. The security of MR4 as a response threshold has also been reported in another study with a cohort of 450 patients [47].

In a cohort of 208 CML patients who had undergone three months of treatment with imatinib, 137 patients (65.8%) achieved CCyR, and 15 (7.2%) achieved MR3, while 11 patients (5.2%) showed any early molecular response. After 12 months, 83.1% achieved CCyR, and 62% achieved MR3, while after a seven-year follow-up, 64.4% reached MR3. The cumulative incidence of MR4 was 51%, and that of MR4.5 was 34.6% after a median time of 3.8 and 5.4 years [48].

The loss of TKI therapy effectiveness and *BCR::ABL1* milestones such as MMR could be supported by the acquisition of TKI-resistant *BCR::ABL1* kinase domain mutations, though once DMR is achieved, it is relatively stable, and the risk of TKI resistance is low [41]. The European LeukemiaNet provides recommendations for *BCR::ABL1* kinase domain mutation analysis, indicating when and how to perform mutation analysis and how to translate results into clinical practice [49]. A Brazilian working group has also provided recommendations for discontinuing tyrosine kinase inhibitors. In Molica et al., 17 out of 208 patients showed fluctuations in *BCR::ABL1* quantification, of which just 2 exhibited a mutation of the *ABL1* kinase domain (E255K and M351T) [48]. In a report with 125 imatinib-resistant CML patients, 28 (22.4%) showed a *BCR::ABL1* kinase domain mutation, from which 7.2% had T315I and 3.2% had E255K [50]. A recent Chinese report evaluated *ABL1* mutation in a sub-cohort of 175 patients who exhibited TKI resistance (first-line TKIs: 164 patients, 93.7%; second-line TKIs: 11 patients, 6.3%) and found that just 54 harbored mutations in the kinase domain, with there being a greater frequency of T315I and E255K [51]. In a Brazilian survey, 48 out of 193 CML patients showed mutations in *ABL1*, with the highest frequencies found for T315I and G250E (20.83% and 14.5%, respectively) [52]. Another Brazilian report showed no *ABL1* mutations in 58 CML patients undergoing treatment with imatinib who showed a suboptimal response [53]. An interesting report suggested that only two-thirds of ABL kinase domain mutations weaken imatinib affinity by more than two-fold compared to the *ABL1* wild type, while, surprisingly, one-third remained sensitive to imatinib and bind with similar or higher affinity than the wild type, identifying three clinical *ABL1* mutations that bind imatinib with wild type-like affinity but with considerably faster dissociation [54].

The treatment-free remission protocol relies on a social-economic benefit in CML clinical management. In one of the first reports in this field, Mahon et al. recruited CML patients ≥ 18 years of age in MR^5^ and monitored 69 patients for at least 12 months. A total of 42 patients relapsed (40 before six months, 1 patient in the seventh month, and 1 in the nineteenth month). At 12 months, the probability of persistent MR^5^ for these 69 patients was 41%. All patients who relapsed responded well to the reintroduction of imatinib, suggesting that imatinib can be safely discontinued in patients with MR^5^ for at least two years [55]. A similar clinical trial was conducted by Ross et al., which included imatinib withdrawal in 40 chronic-phase CML patients. The rate of stable treatment-free remission was 47.1% at 24 months, and most relapses occurred within 4 months after stopping imatinib. No relapses were observed beyond 27 months, and all patients who relapsed remained sensitive to imatinib reintroduction, demonstrating the safety and efficacy of imatinib withdrawal in selected patients [56]. A long-term study enrolled 40 imatinib-treated patients with undetectable *BCR::ABL1* mRNA levels (approximately MR^4.5^) under TFR protocol and with a median follow-up of 8.6 years (ranging from 5.7 to 11.2 years). Eighteen patients remained in continuous TFR, and no patient progressed to the advanced phase, suggesting long-term safety and remarkable stability of response after imatinib discontinuation in appropriately selected CML patients [57]. A recent report by Goni et al. recruited 26 CML patients on generic imatinib for ≥3 years and in sustained deep molecular response, where the median follow-up was 33 months, and 42.3% continued to be in TFR, with all patients who restarted on generic imatinib regaining a major molecular response [58]. A similar approach in 190 CML Chinese patients enrolled in the TFR protocol showed a success rate of 76.9% (95% CI, 70.2–82.4%), 68.8% (95% CI, 61.3–75.2%), and 65.5% (95% CI, 57.4–72.5%) at 6, 12, and 24 months after stopping TKI, and 98.2% of patients who needed to restart TKI treatment quickly achieved MMR [59]. In a retrospective analysis of 168 CML patients recruited for the TFR protocol, 112 of the patients were treated with imatinib and 56 with second-generation TKIs, and 73.2% maintained MR^4^. The median time from TKI discontinuation and the loss of MMR to TKI treatment restarting in the TFR failure group was 4 months and 4.4 months, respectively. This report suggests a threshold value of *BCR::ABL1* RNA expression for TKI discontinuation, where an absolute value of <0.0051% at six months was associated with an extremely high chance (over 90%) of maintaining MMR after treatment discontinuation, both for patients stopping imatinib or 2G-TKI [60]. This result corroborates a Canadian report showing that a shorter *BCR::ABL1* doubling time was associated with a higher rate of TFR failure [61].

The molecular detection of *BCR::ABL1* and *ABL1* mutations is consolidated with real-time PCR. However, digital PCR methods are emerging, even for *BCR::ABL1* quantification and for subsequent TKI discontinuation [62,63,64]. A report by Chung et al. evaluated the performance of the QXDx *BCR::ABL1* %IS (Bio-Rad, Hercules, CA, USA) droplet digital PCR (ddPCR) assay—the first commercially available tool of its type—and found a very strong correlation between this assay and the real-time PCR ipsogen *BCR::ABL1* Mbcr IS-MMR (Qiagen, Hilden, Germany), suggesting that dPCR can provide reliable results for CML patients [65]. A more recent report on the determination of the analytical and clinical performance of dPCR assay showed a limit of detection of MR4.7 (0.002%) and a linear range of MR0.3–4.7 (50–0.002%IS), suggesting that the assay is an accurate, precise, and sensitive system for the diagnosis and monitoring of CML [66]. About *BCR::ABL1* analysis until the moment no divergence results could be identified, as shown in Table 6. However, in the mutations analysis, we could detect divergent results, suggesting that the use of dPCR for monitoring *ABL1* mutations has a great amount of influence on the prognosis.

In this manuscript, we describe the molecular management in a single-center Brazilian CML cohort, especially with respect to *BCR::ABL1* quantification using an in-house duplex one-step RT-qPCR protocol and detecting *ABL1* mutations. As suggested by international guidelines, the molecular monitoring of CML patients provides clinical benefits, allowing clinicians to make assertive and correct decisions regarding TKI choice and the proper follow-up of patients recruited for TFR protocol. Beyond the clinical benefits, the molecular monitoring of CML patients provides socio-economic advantages that warrant its recommendation by the Brazilian public health system. This manuscript describes and discusses the molecular biology usage in CML patients as an advance in clinical practice using known biomarkers such *BCR::ABL1* expression and *ABL1* mutations. An interesting point left out in this manuscript relies on novel genetic alterations in CML patients that may be represent risk factors for disease progression and treatment response. This topic is widely studied worldwide by several groups, including our research group, in various scientific fields to access information that contributes to the determination and validation of novel biomarkers for these patients.

In conclusion, we demonstrate that molecular biology for *BCR::ABL1* quantification and *ABL1* mutation screening represents an essential and necessary approach for CML treatment follow-up and therapy adaptation. The suggested *BCR::ABL1* RT-qPCR quantification protocol [13] for CML patients relies on periodical molecular monitoring, wherein we look for expression levels that could indicate a therapeutical failure, taking to a TKI switch. In this situation, *ABL1* mutation screening should be performed to assertively determine the mutation to select the optimal TKI. For this, the digital PCR approach described in this paper is a methodology that we proposed to be used especially for *ABL1* mutations when *BCR::ABL1* expression shows signs of increased expression, anticipating the *ABL1* mutation and performing the TKI change.

## 4. Methods

Study cohort—clinical samples from patients with chronic myeloid leukemia were collected from diagnostic and confirmed cases in Erasto Gaertner Hospital, Curitiba, Paraná, for 33 months from May 2020 until February 2023 following approval from the Erasto Gaertner Hospital Ethics Committee (CAAE 08809419.0.0000.0098). Sample collection and experimental design followed Brazilian guidelines and regulations. The project was described in detail for all participants who read, discussed, and signed an informed consent form before sample collection. For each patient, 4 mL of peripheral blood was collected in EDTA tubes and processed within 24 h. Patients could have blood collected several times during treatment and clinical follow-up during the project period. This means that mostly recruited patients had more than one sample collected, resulting in a much greater amount samples than patients analyzed. The blood was centrifuged (500× g/10 min/4 °C), and the buffy coat was collected with a Pasteur pipet in a 15 mL tube. RNA extraction was processed utilizing a Qiamp RNA Blood Mini Kit (Qiagen, Hilden, Germany), while DNA was extracted with a Qiamp DNA Blood Mini Kit (Qiagen, Hilden, Germany), both by following the instruction manual. Personal and clinical data such as age, gender, diagnostic date, and treatment were also accessed from patients’ charts.

### BCR::ABL Quantification

Standard curves—for the *BCR::ABL1* target, a partial sequence from the fusion transcript b3-a2 was purchased from GenScript (Piscataway, NJ, USA), where it was cloned in pGEM^®^-T Easy Vector Systems (Promega, Madison, WI, USA). For *ABL1*, RNA from the Jurkat cell line was extracted with Qiamp RNA mini kit (Hilden, Germany) and synthesized to cDNA with Random Primers and SuperScriptIII (ThermoFisher, Waltham, MA, USA). After that, using *ABL1* primers, an end-point PCR was performed using Platinum™ Taq DNA Polymerase High Fidelity (ThermoFisher, Waltham, MA, USA) and PCR product purified with QIAquick PCR Purification (Qiagen, Hilden, Germany). The purified amplicon was cloned in pGEM^®^-T Easy vector (Promega, Madison, WI, USA). Both *BCR::ABL1* p210 and *ABL1* vectors were transformed into *Escherichia coli* cells grown in LB-Amp medium for 16–20 h, and plasmid DNA was extracted with a GeneJET kit (ThermoFisher, MA, USA). The plasmids were measured in NanoDrop One (ThermoFisher, MA, USA), and concentrations were transformed into copy numbers using the following formula: Copy number = (mass × 6.022 × 10^23^)/(length × 1 × 10^9^ × 650), where 6.022 × 10^23^ is Avogadro’s number and 650 is the average weight of a base pair.

*BCR::ABL1* quantification assay—after leukocyte RNA extraction, the molecular quantification of *BCR::ABL1* was made using an in-house one-step duplex qPCR. In this assay, two sets of primers were used simultaneously. The first set was designed to detect both *BCR::ABL1* b3-a2 and b2-a2 isoforms with a probe labeled with HEX, and the second detected *ABL1* in both *BCR::ABL1* b3-a2 and b2-a2 isoforms and wild-type transcript, with a probe labeled with FAM. All primers and probes were purchased from IDT (Coralville, IA, USA) and resuspended in nuclease-free water (ThermoFisher, Waltham, MA, USA). The molecular quantification of *BCR::ABL1* was performed in a final volume of 20uL using TaqPath™ 1-Step RT-qPCR Master Mix, CG (ThermoFisher, Waltham, MA, USA), 2.4uL of oligonucleotides, and 5uL of RNA in LightCycler96 equipment (Roche, Basel, Switzerland). The reaction conditions were 50 °C for 30 min, 95 °C for 5 min, and 45 cycles of 95 °C for 15 s and 60 °C for 30 s. Linear regressions of the *BCR::ABL1* and *ABL1* standard curves were generated by plotting concentration and cycle threshold on the *x* and *y* axes, respectively, generating slope, Y-intercept, and r^2^ values. These standard curve parameters were used to assess the absolute concentrations of *BCR::ABL1* and *ABL1* on the secondary reference calibrator panel for each sample, and a ratio of BCR-ABL/*ABL1* was calculated. The laboratory-specific conversion factors were generated using these data. Values were used for each clinical sample’s *BCR::ABL1* quantification.

Limit of quantification—nine independent experiments with a standard curve of five points with ten-fold dilution in duplicate were used, ranging from 55 × 10^6^ copies/well to 5 × 10^2^ copies/well.

Amplification efficiency and repeatability—to access amplification efficiency without using the standard curve but with a real scenario, we used RNA extracted from the KCL-22 cell line, which is t(9;22) positive. Three independent experiments were performed, using five points with two-fold dilution in duplicate, from 200 ng until 12.5 ng of RNA. This assay was used to calculate repeatability parameters, which was determined as relative standard deviation percentage in repeatability conditions (rRSD%), calculated by the formula rRSD% = repetitions SD × 100/average of repetitions.

Analytical specificity and sensibility—200 ng of extracted RNA from 16 human cell lines were used, of which 14 were negative and 2 were positive for t(9;22) expression: A549 (lung); Capan, MiaPaCa, Panc1, and AsPC1 (Pancreas); MCF7 and MDA-MB-231 (breast); Huh7.5 (liver); SW20 (colon); HL60, THP1, ReH, and Jurkat (hematopoietic); NHDF (fibroblast); K562 and KCL-22 (chronic myeloid leukemia).

Cross-validation assay—two approaches were evaluated to compare *BCR::ABL1* quantification: First, 24 samples were quantified in parallel by a private laboratory. Second, to avoid sampling, equipment, and manipulation bias, 19 samples were quantified in our laboratory using the commercially available Kit XGEN MIX p210 (Mobius Life Science, Pinhais, Paraná, Brazil).

Assay validation and certification—a secondary reference calibrator panel was purchased from Dr. Michele Bianchini of CIO-FUCA, Alexander Fleming Institute, Buenos Aires, Argentina. This secondary reference calibrator panel is composed of a set of five vials in duplicate, each one containing a mixture of lyophilized K562 and HL-60 cell lines in different proportions, reproducing different percentages of *BCR::ABL1* in a ratio of *BCR::ABL1*/*ABL1* between 10% and 0.001%. This secondary reference calibrator panel was necessary for *BCR::ABL1* quantification on the International Scale (IS), deriving laboratory-specific conversion factors (CF). The panel could also determine the assay sensitivity of the established TKI clinical response criteria between 10%, 1%, 0.1% (MR^3^), and 0.01% (MR^4^). To give a deeper assay performance, the secondary reference calibrator panel presented an additional dilution (0.001%, or MR^5^) to assess the assay limit of detection; however, it was not considered for the estimation of the CF. Each panel duplicate had RNA extracted on different days using the RNeasy mini kit (Qiagen) and following the instructions. The *BCR::ABL1* quantification assay was performed four times on different days, with the standard curve ranging from 5 × 10^6^ to 5 × 10^1^
*BCR::ABL1* and *ABL1* copies per well. The BCR-ABL/*ABL1* ratio was calculated for all assays to calculate the CF.

Digital PCR—digital PCR was performed to assess information about the molecular response of patients in treatment-free remission (TFR) or in patients in, at least, MR^3^ or to assess information about mutation in *ABL1*. For this purpose, two different master mixes were purchased from Qiagen—Qiacuity OneStep Advanced Probe kit and Qiacuity UCP Probe PCR kit—which were used as recommended. To detect *ABL1* mutations, *rs121913459* (T315I), *rs121913448* (E255K), and *rs121913452* (V359F) were used in TaqMan assays (ThermoFisher) in the concentration recommended in the Qiacuity UCP Probe PCR kit manual. The PCR reactions were amplified, and the images were captured by Qiacuity one using Qiacuity Nanoplate 26k 24-well. The analyses were performed in Qiacuity Software Suite. The samples for the detection of *ABL1* mutations were chosen based on the qPCR genotyping analyses that were performed before dPCR. We chose samples before the detection of the mutation by qPCR, as well as the point where the mutation was detected. Regarding the *BCR::ABL1* analysis, the RNA samples were diluted to 10 ng/µL and used 100 ng in the reaction. For the analysis of *ABL1* mutations, the DNA samples were diluted to 10 ng/µL and used 100 ng in the reaction as well.

*ABL1* mutation—the following *ABL1* mutations were screened in CML patients: *rs121913459* (T315I), *rs121913448* (E255K), *rs121913461* (Y253H), *rs121913449* (E255V), and *rs121913452* (V359F) using TaqMan Genotyping assays (ThermoFisher), TaqPath ProAmp Master Mix (ThermoFisher, Waltham, MA, USA), and 50ng of DNA in QuantStudio 5 (ThermoFisher, Waltham, MA, USA). These same mutations were evaluated in some CML patients by digital PCR, as described above.

## Figures and Tables

**Figure 1 ijms-24-10118-f001:**
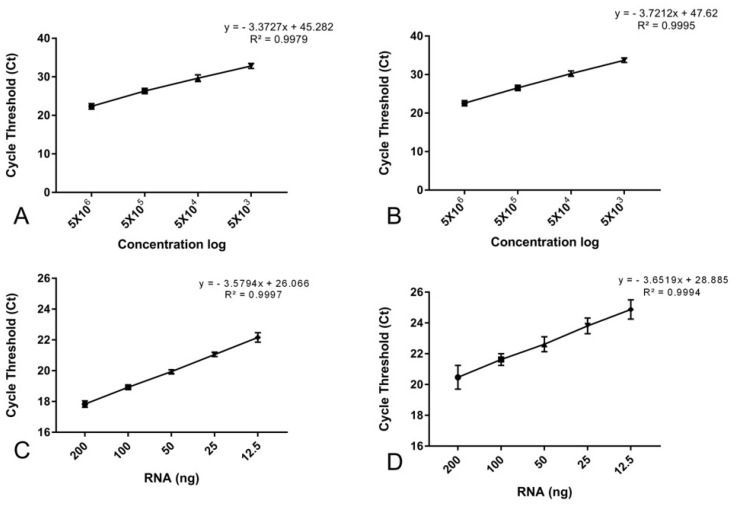
Linear regression of limit of quantification for *BCR::ABL1* (**A**) and *ABL1* (**B**) demonstrating that 500 copies for both targets show an r^2^ > 0.99. Linear regression for *BCR::ABL1* (**C**) and *ABL1* (**D**) quantification using RNA from KCL-22 cell line standard curve, where amplification efficiency was 90% and 87%, respectively.

**Figure 2 ijms-24-10118-f002:**
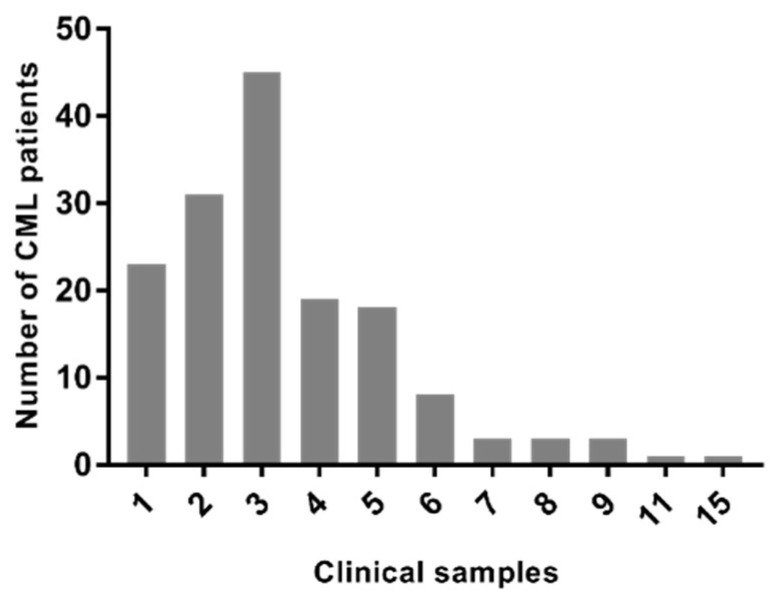
CML clinical samples per patients, indicating that most patients had three and two clinical samples, while few patients had six or more.

**Figure 3 ijms-24-10118-f003:**
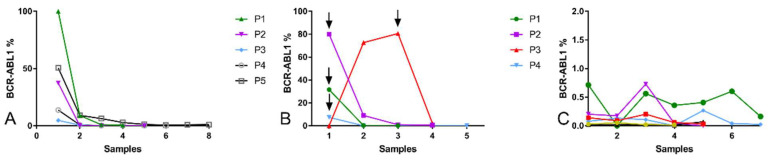
(**A**) *BCR::ABL1* quantification in CML patients that respond well to TKI treatment, as shown by it decreasing levels according to the sequence of analyzed samples. (**B**) *BCR::ABL1* quantification in patients requiring a change in the TKI used. The arrow represents the last clinical samples before imatinib was discontinued. (**C**) Refractory CML patients where *BCR::ABL1* expression are oscillating at a low level, where each color line represents one patient.

**Figure 4 ijms-24-10118-f004:**
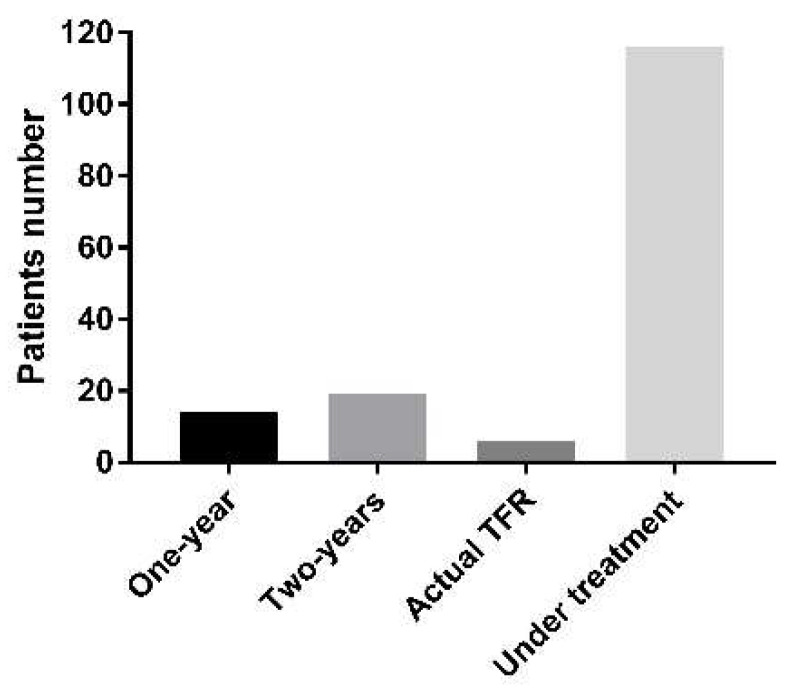
CML patients according to TFR distribution: 6 patients are actual under TFR—3.8%; 14 patients are in DMR or undetected *BCR::ABL1* expression in one-year follow-up—9%; 19 patients presents TFR requirements and eligible for imatinib discontinuation—12.2%; 116 patients are under TKIs treatment and not eligible for TFR—75%.

**Figure 5 ijms-24-10118-f005:**
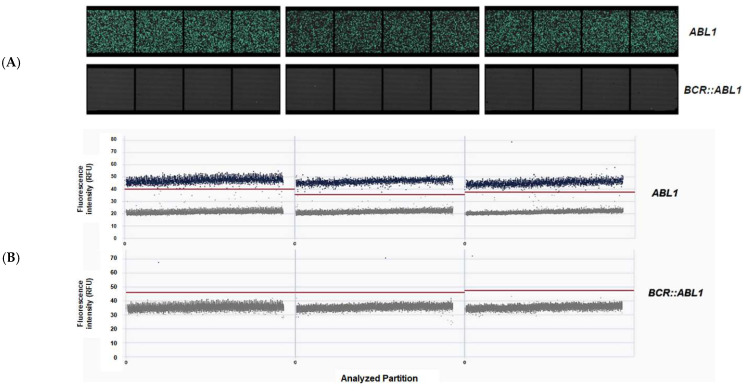
Representative images from dPCR detection of *BCR::ABL1* in three samples. (**A**) Signalmap of *ABL1* and *BCR::ABL1*. Each green spot represents one of 26,000 partitions per well, in which amplification occurred; (**B**) Scatterplot. In the abscissa are the analyzed partitions, where each gray or blue spot represents one of 26,000 partitions per well (scale 0 to 26,000). The red line is the threshold that is automatically set by the analysis software QIAcuity Software Suite v2.1.7.182. The spots below this line (gray) are from the negative partitions, while the spots above (blue) represent the positives partitions for the target.

**Figure 6 ijms-24-10118-f006:**
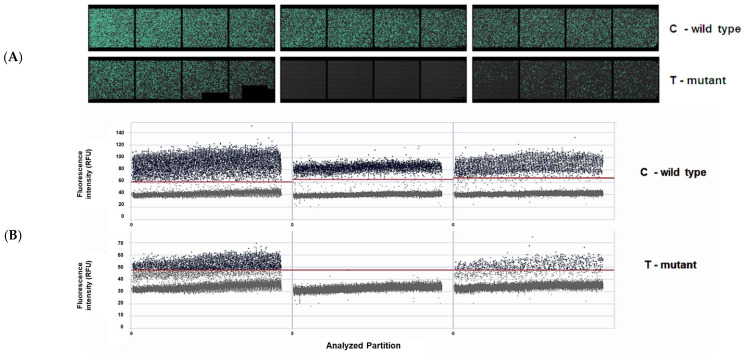
Representative images from dPCR detection of T315I from three samples. (**A**) Signalmap of C (wild type) and T (mutant) bases. Each green spot represents one of 26,000 partitions per well, in which amplification occurred: (**B**) Scatterplot. In the abscissa are the analyzed partitions, where each gray or blue spot represents one of 26,000 partitions per well (scale 0 to 26,000). The red line is the threshold that is automatically set by the analysis software QIAcuity Software Suite v2.1.7.182. The spots below this line are from the negative partitions, while the spots above represent the positives partitions for the target.

**Figure 7 ijms-24-10118-f007:**
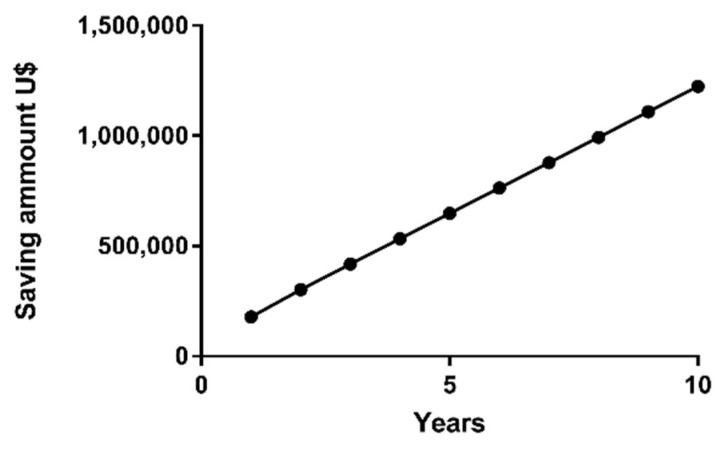
Cost-saving perspective with molecular BCR-ABL quantification in TFR patients for 10 years in Erasto Gaertner Hospital, reaching about $650,000.00 in 5 years and $1,200,000.00 in 10 years.

**Table 1 ijms-24-10118-t001:** Linear regression for *BCR::ABL1* (left panel) and *ABL1* (right panel) quantification using RNA from KCL-22 cell line standard curve, where amplification efficiency was 90% and 87%, respectively.

	*BCR::ABL1*		*ABL1*	
RNA (ng)	Ct (SD)	rRSD%	Ct (SD)	rRSD%
200	17.84 (0.207)	1.16	20.47 (0.771)	3.766
100	18.92 (0.063)	0.333	21.62 (0.378)	1.748
50	19.94 (0.116)	0.582	22.61 (0.484)	2.141
25	21.05 (0.136)	0.646	23.81 (0.505)	2.121
12.5	22.16 (0.307)	1.385	24.87 (0.626)	2.517

**Table 2 ijms-24-10118-t002:** Quantification of *BCR::ABL1* in CML samples using DualQuant and the parallel results for the same samples provided by a private laboratory (reference). Samples with *BCR::ABL1* detection lower than the quantification limit are represented by “Below LoQ”.

Sample	*BCR::ABL1* %	
	DualQuant	Reference
1	ND	Below LoQ
2	0.0285	0.0064
3	0.0642	0.0112
4	0.0404	0.0155
5	0.0199	0.0158
6	0.0189	0.017
7	0.0607	0.0284
8	0.0659	0.0313
9	0.1409	0.0375
10	0.1166	0.0593
11	0.074	0.108
12	0.4565	0.169
13	0.7138	0.2744
14	1.3648	0.7119
15	8.4595	9.6036
16	Below LoQ	Below LoQ
17	ND	Below LoQ
18	Below LoQ	Below LoQ
19	Below LoQ	ND
20	ND	ND
21	ND	ND
22	0.0428	ND
23	0.7309	ND
24	ND	ND

ND: Not detected.

**Table 3 ijms-24-10118-t003:** Quantification of *BCR::ABL1* in CML samples using DualQuant and Mobius commercial kit. Samples with *BCR::ABL1* detection but lower than the quantification limit are represented by “Below LoQ”.

	DualQuant			Mobius		
Samples	*BCR::ABL1* Ct	*ABL1* Ct	%	*BCR::ABL1* Ct	*ABL1* Ct	%
1	28.79	21.87	0.074	31.52	23.11	0.085
2	31.44	24.14	0.066	34.69	24.83	Below LoQ
3	32.37	24.28	0.040	35.28	24.98	Below LoQ
4	29.27	21.65	0.061	32.08	22.6	0.041
5	32.1	23.14	0.019	35.01	24.15	Below LoQ
6	ND	25.75	NA	37.97	26.59	Below LoQ
7	24.08	23.8	8.459	26.35	24.83	9.377
8	ND	21.77	NA	ND	22.67	NA
9	ND	23.8	NA	ND	24.88	NA
10	28.4	22.64	0.457	31.15	24.12	0.218
11	34.78	26.03	Below LoQ	36.63	24.13	Below LoQ
12	33.96	29.75	Below LoQ	ND	37.98	NA
13	30	25.28	0.714	32.55	26.05	0.309
14	28.16	24.46	1.365	29.17	24.56	1.136
15	31.39	23.74	0.141	32.91	24.07	0.063
16	28.84	22.66	0.117	32.35	23.64	0.069
17	31.27	22.75	0.029	34.87	23.85	Below LoQ
18	ND	23.35	NA	ND	24.37	NA
19	24.96	25.05	5.890	26.82	24.23	4.531

ND: Not detected. NA: not available.

**Table 4 ijms-24-10118-t004:** Number and percentages of patients in each level of response to TKIs by year of follow-up. DMR refers to the sum of patients with MMR4.0 or less (sum of MMR 4.0, MMR 4.5, MMR 5, and undetectable levels of BCR-ABL IS%).

Year	Unique Patients (n)	%IS > 1 n (%)	CCR (IS ≤ 1%) n (%)	MMR 3 (IS ≤ 0.1%) n (%)	MMR 4 (IS ≤ 0.01%) n (%)	MMR 4.5 (IS ≤ 0.0032%) n (%)	MMR 5 (IS ≤ 0.001%) or Undetectable n (%)	DMR (≤MMR 4.0) n (%)
2020	60	11 (18.3%)	7 (11.7%)	16 (26.7%)	5 (6.7%)	0 (0%)	22 (36.6%)	26 (43.3%)
2021	124	17 (13.7%)	16 (12.9%)	30 (24.2%)	4 (3.2%)	0 (0%)	57 (46%)	61 (49.2%)
2022	126	14 (11.1%)	10 (7.9%)	33 (26.2%)	9 (7.1%)	2 (1.6%)	58 (46.1%)	69 (54.7%)
2023	23	5 (21.7%)	3 (13%)	11 (47.8%)	0 (0%)	0 (0%)	4 (17.4%)	4 (17.4%)
Total (n)	334	47	36	90	18	2	141	161

**Table 5 ijms-24-10118-t005:** Number and percentages of patients in each level of DMR in the 33 months of follow-up. DMR refers to the sum of patients with MMR4.0 or less (sum of MMR 4.0, MMR 4.5, MMR 5, and undetectable levels of BCR-ABL IS%).

2020–2023	IS > 1%	CCR (<1%)	MMR 3 (<0.1%)	MMR 4.0 (<0.01%)	MMR4.5 (<0.0032%)	MMR 5.0 (<0.001%) or ND	DMR (≤MMR 4.0)
n	17	17	45	13	1	85	99
%	9.5%	9.5%	25.3%	7.3%	0.6%	47.8%	55.6%

**Table 6 ijms-24-10118-t006:** Comparison of results obtained by dPCR and qPCR (%IS) from 22 patients that are in DMR, where * indicate patients that showed no qPCR BCR::ABL detection but with copies detected by dPCR and ^#^ patients with BCR::ABL quantification by qPCR and no detection in dPCR.

	*BCR::ABL1*	
	dPCR Copies/Reaction	qPCR %IS
Patient 1 *	0.188	0
Patient 2	0.196	0.009
Patient 3	0	0
Patient 4	0	0
Patient 5	0	0
Patient 6 *	0.2	0
Patient 7	0	0
Patient 8 *	0.208	0
Patient 9	0.204	0.065
Patient 10 *	0.2	0
Patient 11 *	0.208	0
Patient 12 *	0.204	0
Patient 13	0	0
Patient 14	0	0
Patient 15 *	0.204	0
Patient 16 ^#^	0	0.008
Patient 17 ^#^	0	0.005
Patient 18 *	0.408	0
Patient 19 *	0.204	0
Patient 20	0.208	0.004
Patient 21	0.2	0
Patient 22	0	0

**Table 7 ijms-24-10118-t007:** Comparison of results obtained by qPCR and by dPCR to investigate *ABL1* mutations.

T315I					
Patient ID	Sample ID	qPCR	dPCR	%*BCR::ABL1* (IS)	Treatment
Patient 1	Sample 1	WT	WT	0.054	Nilotinib
	Sample 2	T315I	T315I	100	
Patient 2	Sample 1	WT	WT	22.92	BMT
	Sample 2	WT	T315I	NA	
Patient 3	Sample 1	WT	T315I	0.281	BMT
Patient 4	Sample 1	WT	T315I	0.315	Dasatinib
Patient 5	Sample 1	WT	T315I	8.9	Dasatinib
Patient 6	Sample 1	WT	T315I	1,38	Imatinib
**E255K**					
**Patient ID**	**Sample ID**	**qPCR**	**dPCR**	**%*BCR::ABL1* (IS)**	**Treatment**
Patient 7	Sample 1	WT	WT	61.12	Imatinib
	Sample 2	WT	E255K	72.7	
	Sample 3	E255K	E255K	80.6	
	Sample 4	WT	WT	8.19	Dasatinib
**V359F**					
**Patient ID**	**Sample ID**	**qPCR**	**dPCR**	**%*BCR::ABL1* (IS)**	**treatment**
Patient 8	Sample 1	WT	WT	47.3	None
	Sample 2	V359F	V359F	43.2	Nilotinib

Wt represents “wild-type”; BMT “Bone marrow transplantation”; and NA “Not available”.

**Table 8 ijms-24-10118-t008:** Table illustrating the cost saving in sub-cohort of TFR eligible patients, showing patients number, monthly, and annual cost, demonstrating a saving of $302,400 in two years just with this cohort.

Group	TFR Patients	TFR Success (65%) ^(a)^	Monthly Cost (USD)	Annual Cost/Patient ^(b)^	Annual Cost (USD) ^(a,b)^
1 (immediately)	19	12	1200	14,400	172,800
2 (In one year)	14	9			129,600
In two years	33	21			302,400

^(a)^ TFR success rate calculation (65% of the total number of TFR patients); ^(b)^ annual cost per patient (monthly cost multiplied by 12); ^(a,b)^ annual cost per patient multiplied by the number of estimated TFR successful patients.

## Data Availability

No new data were created.

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
