# Peer review of "Molecular BCR::ABL1 Quantification and ABL1 Mutation Detection as Essential Tools for the Clinical Management of Chronic Myeloid Leukemia Patients: Results from a Brazilian Single-Center Study"

_ijms, 2023, doi:10.3390/ijms241210118_

Round 1

Reviewer 1 Report

Comments are included in the attached file.

Because I am not a native speaker I cannot accurately assess the grammar and punctuation of the English language. From my point of view, the work is written clearly and is readable

Author Response

The authors thank you the reviewer for the relevant contribution and find below the answers about the points. We added a clean version, where we highlighted the alterations made on the manuscript. 

  1. The Introduction section is unusually long. The data in Table 1, which are not the authors' original findings, can be described in the text or summarized in the schematic Figure. I propose to move some parts of the Introduction to the Discussion section. Considerations of the economic implications of introducing molecular detection of the BCR::ALB1 mutation and its impact on the cost of treatment are not a molecular science topic. Perhaps it would be more logical to place them as supplementary files.

Answer: Introduction section was reduced as highlighted in yellow parts. The table I was removed and the following sentence was inserted and highlighted in green “as patietns harboring T315I mutation are recommended use of Ponatibib, while V299L Nilotinib or ponatinib and Y253H, E255V/K, F359V/I/C Dasatinib, bosutinib or ponatinib”. Regarding economic implications in introduction, the authors considered it as a relevant topic on this manuscript, however as you said is not a direct molecular science topic but is correlated with patients and social relevance, especially in a medium income economical country like Brazil. This topic was addressed in relation to Brazilian CML and discussed on the study cohort later in the article.

  1. The Materials and Methods section would be clearer if it were divided into subsections.

Answer: The Methods section was divided into 2.1. Study cohort; 2.2 BCR::ABL Quantification; 2.3 Digital PCR; 2.4 ABL1 mutation. The section 2.2 was sub-divided into 2.2.1 Standard curve; 2.2.2 BCR::ABL1 quantification assay; 2.2.3 Limit of quantification; 2.2.4 Amplification efficiency and repeatability; 2.2.5 Analytical specificity and sensibility; 2.2.6 Cross-validation assay; 2.2.7 Assay validation and certification;

  1. In the Results section, I suggest to concentrate Figures 1 and 2 as well as Tables 2, 3, 4 into a maximum of two documentary outputs.

Answer: Figures 1 and 2 concentrated in one, now refered as figure 1. Regarding tables 2, 3 and 4, the authors belive that merge then into one or two tables is not possible because they are related to different results and could generate confusing tables.

  1. Please summarize the data in Figure 3 in the definition of the patient cohort in the Material and Methods section.

Answer: Following reviewer suggestion, the authors added the following statment highlighted in red in 2.1 Study cohort: Patients could have blood collected several times during treatment and clinical follow-up during the project period. This means that mostly recruited patients had more than one samples collected, resulting in much more samples than patients analyzed.

  1. Figure 4 should include panels A, B, and C, but these are not shown.

Answer: Thank you for the note, letters for panels A, B and C inserted.

  1. The pie chart in Figure 5 is not an optimal way of documenting such data; a table or bar chart seems better to me.

Answer: Graphic changed to bar chart.

  1. Figures 5 and 7 are low contrast, which makes them difficult to understand. What are the values that are on the abscissae? They start with the number 0, apart from which there are no other numerical data. In the description below the abscissa is the label "analyzed partition", this also needs to be explained.

Answer: Now are figures 5 and 6, and the image resolution has been improved. In the abscissa axel is the total of partitions that were used to the analysis. Despite each well contains 26.000 partitions is very rare that we have amplification product in all of them. Then, the software give us the number of valid partitions and the not valid partitions, and the number of copies is calculated considering the valid partitions. By this reason, this scale in abscissa axel are not appointed in the figure. This is a figure directly provided by Qiagen dPCR instrument. Legend was updated, as highlighted in yellow.

  1. I very much appreciate the effort to detect BCR::ALB1 mutations using dPCR. However, I think the experimental paper should combine these data with sequencing data of the corresponding PCR products. The dPCR method is recommended to detect the expected mutations. However, using only this method deprives the authors of the opportunity to capture previously undescribed mutations. These should be addressed at least in Discussion.

Answer: The dPCR used on this manuscript in fact is to detect specific mutations at ABL1, not to discover novel ones. Our proposal is to use molecular biology for BCR::ABL1 quantification and ABL1 mutations detection to be directly used in Brazilian CML patients clinical practice. We used and highlighted dPCR as a sensitive, specific and which provides an absolute quantification of main ABL1 mutations. Following reviewer sugestion, we add the following paragraph in the end of discussion, highlighted in yellow “This manuscript describes and discuss the molecular biology usage in CML patients as an advance in clinical practice using known biomarkers such BCR::ABL1 expression and ABL1 mutations. An interesting point left out in this manuscript relies on novel genetic alterations in CML patients that may be represent risk factor for the disease progression and treatment response. This topic is widely studied worldwide by several groups, including our, on basic sciences field to access information that contribute to determinate and validate novel biomarkers for these patients.”

  1. I would have appreciated if the authors had included a Conclusion chapter where they would have provided their suggestion on how to approach the diagnosis of newly diagnosed CML patients and, during their treatment, how to modify treatment options according to the data obtained. For this purpose, they may draw up a schematic Figure

Answer: Thank you for this point. On this manuscript we did not propose protocols for CML diagnostic or monitoring, where it was demonstrated it importance in clinical practice. So, as requested we added a last paragraph  as conclusion, highlighted in light blue.

Reviewer 2 Report

The manuscript presents the results on the role of BCR::ABL1 in chronic myeloid leukemia. This is an interesting results, however, there are several issues that should be addressed:

1. The title should reflect the content of the manuscript. Since authors mostly described BCR:ABL it should be stated in the title.

2. Method section should be subdivided in the subsections describing each method separately 

3. A lot of sentences are missing references throughout the manuscript 

4. Authors should describe immunological aspects of CML in the Introduction

5. It is unclear whether newly diagnosed patients or relapsed patients participated in the study.

6. Authors also haven't described any new therapies ( immunotherapies?) used for the treatment of CML

Author Response

The authors thank you the reviewer for the relevant contribution and find below the answers about the points. We added a clean version, where we highlighted the alterations made on the manuscript. 

  1. The title should reflect the content of the manuscript. Since authors mostly described BCR:ABL it should be stated in the title.

Answer: Following reviewer suggestion the title was changed to “Molecular BCR::ABL1 quantification and ABL1 mutation detection as essential tools for the clinical management of chronic myeloid leukemia patients: results from a Brazilian single-center study”

  1. Method section should be subdivided in the subsections describing each method separately.

Answer: The Methods section was divided into 2.1. Study cohort; 2.2 BCR::ABL Quantification; 2.3 Digital PCR; 2.4 ABL1 mutation. The section 2.2 was sub-divided into 2.2.1 Standard curve; 2.2.2 BCR::ABL1 quantification assay; 2.2.3 Limit of quantification; 2.2.4 Amplification efficiency and repeatability; 2.2.5 Analytical specificity and sensibility; 2.2.6 Cross-validation assay; 2.2.7 Assay validation and certification;

  1. A lot of sentences are missing references throughout the manuscript 

Answer: References added with references highlighted in gray throughout the manuscript 

  1. Authors should describe immunological aspects of CML in the Introduction

Answer: A paragraph highlighted in yellow was inserted in the end of introduction, as follow: ”CML is characterized by a period of innate and adaptive immune dysfunction at diagnosis, which prevents antileukemia immune responses. Immune suppressor cells including myeloid-derived suppressor cells (MDSCs) and regulatory T-cells (Tregs) contribute to T-cell dysfunction and disease progression in CML, expanding at diagnosis, and reducing following TKI therapy. Clinical data showed that imatinib or dasatinib treated patients exhibit expansion of CD8+ CTLs or NK cells which are associated with an improved response to therapy. Concerning immunotherapies for CML, IFN-α may provide an alternative approach in selected TKI-resistant mutation-positive CML patients, particularly those ineligible for bone marrow transplantation. A combined therapy with Chimeric Antigen Receptor T-cells therapy (CAR-T) and TKI may confer particular therapeutic benefit to TKI-resistant/intolerant, young or advanced phase CML patients

  1. It is unclear whether newly diagnosed patients or relapsed patients participated in the study.

Answer: As stated in “2. Methods - Study cohort — clinical samples from patients with chronic myeloid leukemia, were collected from diagnostic and confirmed cases in Erasto Gaertner Hospital, Curitiba, Paraná”. So we had recruited patients both newly diagnosed patients, relapsed and under treatment patients.

  1. Authors also haven't described any new therapies (immunotherapies?) used for the treatment of CML.

Answer: This topic was added in the introduction together with immunological aspects of CML

Round 2

Reviewer 2 Report

The authors addressed all the comments